# Acylated Flavone *O*-Glucuronides from the Aerial Parts of *Nepeta curviflora*

**DOI:** 10.3390/molecules25173782

**Published:** 2020-08-20

**Authors:** Maysaa Rabee, Øyvind Moksheim Andersen, Torgils Fossen, Kjersti Hasle Enerstvedt, Hijazi Abu Ali, Saleh Rayyan

**Affiliations:** 1Department of Chemistry, Birzeit University, Birzeit 627, Palestine; maysa.rabee@hotmail.com (M.R.); habuali@birzeit.edu (H.A.A.); 2Department of Chemistry and Centre for Pharmacy, University of Bergen, Allégt. 41, N-5007 Bergen, Norway; oyvind.andersen@uib.no (Ø.M.A.); torgils.fossen@uib.no (T.F.); kjersti.enerstvedt@uib.no (K.H.E.)

**Keywords:** syrian catnip, *Nepeta curviflora*, lamiaceae, acylated flavone 7-*O*-glucuronosides

## Abstract

*Nepeta curviflora* Boiss. (Syrian catnip) is native to the Middle East. This medicinal plant is commonly used against nervous disorders, rheumatic pains, and high blood pressure. Herbal infusions prepared from various *Nepeta* spp. are extensively consumed as functional food. However, limited information has been known about the phenolic constituents of Syrian catnip. In this study, two acylated flavone 7-*O*-glucuronides, apigenin 7-*O*-(2″-*O*-(2‴-(*E*-caffeoyl)-β-glucuronopyranosyl)-β-glucuronopyranoside) (**1**) and luteolin 7-*O*-(2″-*O*-(2‴-(*E*-caffeoyl)-β-glucuronopyranosyl)-β-glucuronopyranoside) (**2**), along with the known phenolic compounds rosmarinic acid, caffeic acid, apigenin, and apigenin 7-*O*-β-glucopyranoside were isolated from the aerial parts of *N. curviflora*. The characterizations of these compounds were based on high-resolution mass spectrometry, UV, and extensive use of multidimensional NMR spectroscopy. The new compounds (**1** and **2**) were identified in the unmodified state and as dimethylesters.

## 1. Introduction

The genus *Nepeta* (Lamiaceae) is widely distributed in Europe, North Africa, North America, India, and Asia, including the Mediterranean countries. It contains around 300 species, some of which are used in traditional medicine [1]. Intake of some *Nepeta* spp. has been associated with positive health effects, including antispasmodic, antiasthmatic, and anti-inflammatory activities, as well as efficiently maintaining and balancing serum lipids [2,3,4,5]. Some species (for instance *Nepeta menthoides*) have also been used as traditional herbal medicine against nervous disorders, rheumatic pains, and high blood pressure [6], while the aqueous extracts of *N. menthoides* have possible benefits in controlling the mood of patients suffering from major depression [7]. As a consequence, herbal infusions prepared from *Nepeta* spp. are nowadays considered as functional food [8].

The potential beneficial pro-health effects associated with the intake of *Nepeta* species have been suggested to be partly attributable to their content of essential oils and phenolic compounds [1,9,10,11,12,13,14,15,16], which have been reported as the major secondary metabolites of this genus [16,17,18]. Among these, rosmarinic acid appears to be the most abundant phenolic compound [1,17,19,20,21,22].

Syrian catnip, *N. curviflora* Boiss., (syn. *Glechoma curviflora* (Boiss.) Kuntze), a medicinal plant native to the Middle East, has been reported to exhibit antioxidant [23], phytotoxic [24] as well as nematicidal activities [25]. While the screened antimicrobial activity of dimethylsulfoxide extract of *N. curviflora* apparently is low [26], however, methanolic extracts of leaves and stem of this plant have shown efficacy against more than 88.8% of the tested microorganisms [26]. Only volatile chemical constituents have previously been reported from *N. curviflora* [24,25,26,27]. Although phenolic compounds of some *Nepeta* spp. have been suggested to be responsible for a wide range of biological activities [20,28,29], the specific characterization of this group of phenolic compounds appears underinvestigated in this genus. The aim of this study was thus to isolate and elucidate individual phenolic constituents of the aerial parts of Syrian catnip.

## 2. Results and Discussion

The plant was identified by Dr. Munir Naser at Birzeit University, and a voucher specimen of *N. curviflora* has been deposited at Al-Quds University Herbarium (accession number Nc2019Lam11) and at the seeds bank of the Union of Agricultural Work Committees (UAWC) (accession number UB-435-19/s).

The HPLC chromatogram of the methanolic extract of the aerial parts (stems, leaves, and flower) of *N. curviflora* recorded at 360 nm showed two major and several minor compounds. This extract was purified by partition against hexane, followed by Amberlite XAD-7 absorption chromatography. The flavonoids in the purified extract were further fractionated by Sephadex LH-20 chromatography, and pure compounds (**1**, **2**, and **4**) were thereafter isolated by preparative HPLC of selected Sephadex LH-20 fractions.

Compounds **3** and **4** were identified as the methylesters of the known phenolic compounds rosmarinic acid and caffeic acid, respectively (Appendix A). The esters were most probably made by the solvent (acidified methanol) during isolation. Compounds **5** and **6** were isolated from the flowers of the plant and identified as apigenin (**5**) and apigenin 7-*O*-β-glucopyranoside (**6**) by UV (Appendix A) and NMR (Appendix A) spectroscopy.

The downfield region of the ^1^H-NMR spectrum of **1** (Appendix A) showed an AA’XX’ system at δ 7.94 (H-2′/6′) and δ 6.95 (H-3′/5′), a one proton singlet at δ 6.85 (H-3) and an AX system at δ 6.76 (d, *J* = 2.2 Hz; H8) and δ 6.39 (d, *J* = 2.2 Hz; H6), in accordance with a 7-*O*-substituted apigenin derivative (Appendix A). Based on HSQC (Appendix A), HMBC (Appendix A), and H2BC (heteronuclear 2-bond correlation) (Appendix A) NMR spectra of **1**, 15 carbon resonances belonging to the aglycone and 9 resonances corresponding to an acyl moiety, were assigned (Table 1). The presence of two glycopyranosyl units was further suggested from both the ^1^H and ^13^C-NMR spectra (Table 1). The relationships between the ^1^H sugar resonances of each sugar unit were assigned by the ^1^H-^1^H COSY experiment (Appendix A), and the corresponding ^13^C resonances were then assigned by the HSQC experiment. The coupling constants (7.5 Hz and 8.2 Hz) for the two anomeric protons and the twelve ^13^C resonances were consistent with two *O*-β-glucuronopyranosyl units [30,31,32,33]. The additional presence of the two singlets at δ 3.63 and δ 3.52 ppm (methoxy groups), which showed cross peaks with the carbonyl groups at δ 169.10 and δ 169.17, respectively, in the HMBC spectrum, suggested the presence of a methyl ester group attached to each of the two glucuronopyranosyl units. The downfield shift of H-2‴ (δ 4.62) of the terminal glucuronopyranosyl unit indicated acyl substitution. The presence of an AMX system at δ 7.03 (*J* = 2.1; H-2⁗), δ 6.97 (*J* = 2.1, 8.5; H-6⁗), and δ 6.76 (*J* = 8.5; H-5⁗), and *trans*-oriented olefinic protons at δ 7.43 (*J* = 15.8 Hz, H-β) and δ 6.24 (*J* = 15.8 Hz, H-α) established the identity of the acyl-group to be (*E*)-caffeoyl. The cross peaks at δ 5.44/162.19 (H-1″/C-7), δ 3.52/101.42 (H-2″/C-1‴), δ 4.88/81.01 (H-1‴/C-2″), and at δ 4.62/165.87 (H-2‴/C=O) in the HMBC spectrum of **1** confirmed the linkages between the aglycone, sugar, and (*E*)-caffeoyl moieties (Figure 1). The high-resolution ESI^+^-MS spectrum of **1** (Appendix A) showed a [M + H]^+^ ion at *m*/*z* 813.1881 corresponding to the empirical formula C_38_H_37_O_20_^+^ (calc. 813.1878 Da) in agreement with the dimethyl ester of apigenin 7-*O*-(2″-*O*-(2‴-(*E*-caffeoyl)-β-glucuronopyranosyl)-β-glucuronopyranoside) (**1**) (Figure 1).

The NMR resonances of **2** were very similar to those of compound **1** (Table 1). However, the main differences were shown in the aromatic region where the ^1^H NMR spectrum of **2** (Appendix A) revealed a 3H AA’X system at δ 7.43 (H-2′, *br*), δ 7.43 (H-6′, *br*), and δ 6.92 (*J* = 8.5 Hz; H-5′), a 1H singlet at δ 6.73 (H-3) and a 2H AX system at δ 6.73 (d, *J* = 2.2 Hz; H8) and δ 6.39 (d, *J* = 2.2 Hz; H6), in accordance with a 7-*O*-glycosylated luteolin derivative. The cross peaks at δ 5.46/162.22 (H-1″/C-7), δ 3.53/101.46 (H-2″/C-1‴), δ 4.89/81.07 (H-1‴/C-2″), and at δ 4.63/165.94 (H-2‴/C=O) in the HMBC spectrum of **2** (Appendix A) confirmed the linkages between the aglycone, sugar, and (*E*)-caffeoyl moieties (Figure 1). The high-resolution ESI^+^–MS spectrum of **2** (Appendix A) showed a [M + H]^+^ ion at *m*/*z* 829.1830 corresponding to the empirical formula C_38_H_37_O_21_^+^ (calc. 829.1827 Da) in agreement with the dimethylester form of luteolin 7-*O*-(2″-*O*-(2‴-(*E*-caffeoyl)-β-glucuronopyranosyl)-β-glucuronopyranoside) (Figure 1).

Previously, we have reported that the free carboxyl group of glucuronyl moieties of flavonoids readily will be esterified with methanol during extraction and isolation processes involving acidified methanol as solvent [30]. This is in accord with the identification of both 1 and 2 as dimethylesters caused by methylesterfication of the two glucuronyl moieties of these flavonoids. Small amounts of parental 1 and 2 without their methylesters were indeed detected by LC-MS analysis of Sephadex LH-20 fractions of the purified extract of *N. curviflora* (Appendix A).

Antibacterial activity was measured using the agar diffusion method. Sephadex LH-20 fractions containing both **1** and **2** dissolved in DMSO did not reveal antibacterial activity. However, the cruder XAD-7 purified material showed some antibacterial activity, suggesting other compounds in the extract to be considered in future antibacterial activity studies.

## 3. Materials and Methods

### 3.1. General

UV-Vis absorption spectra were recorded on-line during HPLC analysis using a photodiode array detector (HP 1050) (Agilent Technologies, Santa Clara, CA, USA). ^1^H (600.13 MHz) and ^13^C (150.90 MHz) NMR spectra were obtained on a Bruker Biospin AV-600 MHz instrument equipped with a TCI ^1^H-^13^C/^15^N CryoProbe (Bruker BioSpin, Zürich, Switzerland), and on a Bruker Biospin AV-850 MHz equipped with a CryoProbe (Bruker BioSpin, Zürich, Switzerland). All experiments were recorded at 298K and the chemical shift values were set relative to the deutero-methyl ^13^C signal and the residual ^1^H signal of the solvent ((CD_3_)_2_SO) at δ 39.6 and δ 2.49, respectively. Low-resolution mass spectra were recorded on a LC-MS system (Agilent Technologies, Santa Clara, CA, USA) consisting of an Agilent 1200 series LC module (binary pump, column compartment/oven, and auto sampler), equipped with an Agilent ZORBAX SB-C_18_ (RRHT 2.1 × 50 mm, 1.8 µm), with an Agilent 6420A mass spectrometer equipped with a triple quadrupole (QqQ configuration) mass analyzer using electrospray ionization (ESI) as detector.

High-resolution mass spectrometry (ESI^+^/TOF), spectra were recorded using a JEOL AccuTOF JMS-T100LC instrument (JEOL, Peabody, USA) in combination with an Agilent Technologies 1200 Series HPLC system. A Zorbax Eclipse-C18 (Agilent Technologies, Santa Clara, CA, USA) (50 × 2.1 mm, length × i.d., 1.8 µm) column was used for separation. Two solvents, A (H_2_O + 0.2 % HCOOH, *v*/*v*) and B (acetonitrile + 0.2 % HCOOH, *v*/*v*), were used for elution. The following solvent compositions were used: 0 min (0 % B), 0–2 min (0 to 10% B, linear gradient), 2–15 min (10 to 80% B, linear gradient). The flow rate was 0.4 mL/min and the temperature was kept at 60 °C.

### 3.2. Isolation of Flavones

The aerial parts of Syrian catnip (stems, leaves, and flowers) were collected in May 2016 from the close surrounding area of Birzeit University, Palestine (Coordinates: Latitude: 31°57′18.76″ N; Longitude: 35°10′30.32″ E). The collected plant material was dried and thereafter stored for approximately one month. The dried aerial part (755 g) was extracted three times over night at 4 °C, with 5L 10% H_2_O in MeOH (*v*/*v*; containing 0.5% trifluoroacetic acid, TFA). The combined extract was concentrated under reduced pressure in order to remove methanol. The nonpolar compounds were removed by partition against hexane. Approximately 1/3 of the resulting aqueous phase was subjected to Amberlite XAD-7 column chromatography. Part of the XAD-7 purified extract (2.34 g) was further purified and separated on a 100 × 5 cm Sephadex LH-20 column using acetonitrile—H_2_O:TFA (10:90:0.2; *v*/*v*) and MeOH—H_2_O:TFA (80:20:0.2; *v*/*v*) as mobile phase with a flow rate of 4 mL/min (Appendix A).

A mixture of compounds **1**, **2**, and **4** was obtained in the combined fractions 25 and 26 achieved by Sephadex LH-20 chromatography (Appendix A). Pure compounds **1**, **2**, and **4** were then isolated from fractions 25 and 26 by preparative HPLC. Pure compound **3** was eluted in Sephadex LH-20 fraction 33 (Appendix A). The isolation of compounds **5** and **6** was based on the extraction of 195 g dried flowers of *N. curviflora* followed by the same purification and separation steps as indicated above. The dried XAD-7 purified extract (2.31 g) was fractionated by Sephadex LH-20 chromatography, and pure **5** and **6** were obtained in fractions 27 and 38, respectively (Appendix A).

### 3.3. Preparative HPLC

Preparative HPLC was performed using a Gilson 321 preparative HPLC equipped with a UV detection (Dionex UltiMate 3000 Variable Wavelength Detector) (Dionex Corporation, Sunnyvale, CA, US). The system was equipped with an Econosil C18 column (250 mm × 22 mm; length × I.D., 5.0 µm; Fortis Technologies Ltd., Neston, UK). The elution protocol consisted of solvents A, H_2_O containing 0.5% TFA (*v*/*v*) and B, acetonitrile containing 0.5% TFA (*v*/*v*). The following gradient was used: 100% A in 0–5 min, 10% B (isocratic elution) for the next 46 min (6–52 min), 20% B (isocratic elution) for the next 12 min (53–65 min), 50% B (isocratic elution) for the next four minutes (66–70 min), and then back to the starting conditions (100% A, isocratic elution) in 4 min (71–75 min). The flow rate was 12.0 mL min^−1^.

### 3.4. Analytical HPLC

Analytical HPLC was performed using Agilent 1100 HPLC system (Agilent Technologies, Santa Clara, CA, USA) equipped with a HP 1050 diode array detector and with an ODS-Hypersil column (20 × 0.5 cm, length × i.d., 5 μm; Supelco, Bellefonte, PA, USA), using the solvents A, H_2_O containing 0.5% TFA (*v*/*v*) and B, acetonitrile containing 0.5% TFA (*v*/*v*). The following gradient (B in A) was used: 10 to 14% B in 0–10 min, 14% B (isocratic elution) for the next 4 min, 14 to 40% B (linear gradient) from 14–32 min, 40% B (isocratic elution) from 32–43 min, followed by a linear gradient 40% B-10% B for 3 min to re-establish the starting conditions. The flow rate was 1.0 mL min^−1^.

### 3.5. Biological Activity

The antibacterial activities of Sephadex LH-20 fractions containing both **1** and **2,** and XAD-7 purified material, were investigated against five gram-positive (*Staphylococcus aureus, Micrococcus luteus, Bacillus subtilis, Enterococcus faecalis*, and *Staphylococcus epidermidis*) and four gram-negative (*Escherichia coli, Klebsiella pneumonia, Proteus mirabilis*, and *Proteus aeruginosa*) bacteria. The antibacterial test was carried out by using the agar diffusion method. The sterile saline was prepared by dissolving 0.5 g of NaCl in 500 mL of water (0.1% of NaCl) before this solution was autoclaved. The single bacterial colonies were dissolved in the sterile saline until the turbidity of the suspended cells reached the McFarland 0.5 standard. The bacterial inocula were spread on the surface of Mueller–Hinton nutrient agar using a sterile cotton swab. Then, the wells (6 mm in diameter) in the agar plate were made by using Sterile glassy borer [34,35]. The samples were dissolved in DMSO in concentrations of 6 mg/mL, and 25 μL of each were added into their respective wells before the plates were incubated at 37 °C for 12–24 h. Gentamycin (G) and Erythromycin (E) were used as positive controls, while DMSO was used as negative control. The activities of the samples were determined by measuring the inhibition zone diameter in millimeter. The results were determined by calculating the average of three trials.

## 4. Conclusions

In this investigation, individual phenolic constituents of the aerial parts of Syrian catnip have been characterized on the basis of extensive spectroscopic analyses. Two new flavonoids (**1** and **2**) along with two known flavonoids (**5** and **6**) and the methylesters of rosmarinic acid (**3**) and caffeic acid (**4**) have been identified. Flavonoids glycosylated with glucuronic acid have previously been reported from several *Nepeta* spp. [36,37,38]. However, the findings of **1** and **2** in *N. curviflora* are the first report of acylated flavone glucuronides in the genus *Nepeta*, which might have chemotaxonomic significance within the genus.

## Figures and Tables

**Figure 1 molecules-25-03782-f001:**
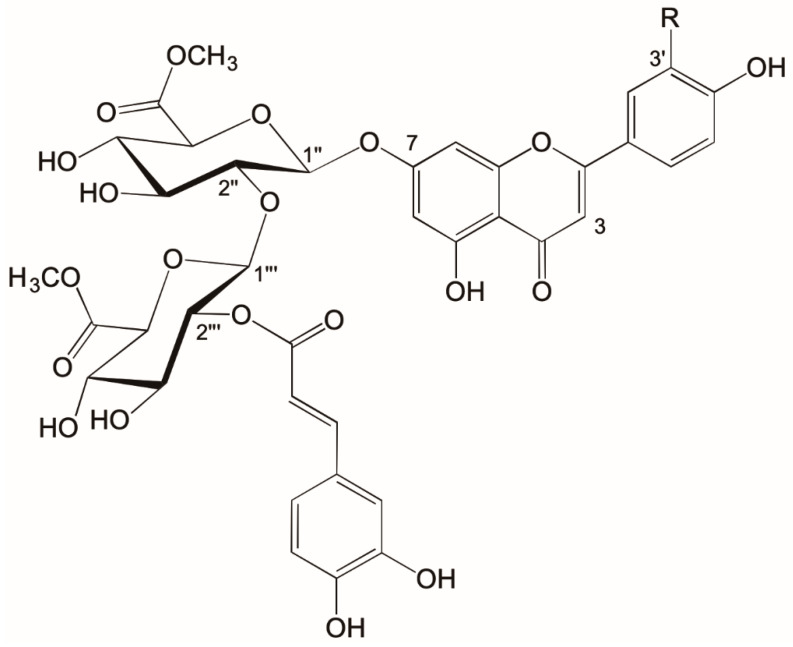
Structures of two new flavonoids **1** (R=H) and **2** (R=OH) isolated from aerial parts of *Nepeta curviflora*.

**Table 1 molecules-25-03782-t001:** ^1^H and ^13^C spectral data (δ in ppm) for **1** and **2** dissolved in DMSO-*d*_6_ at 25 °C.

	1 (^1^H)	1 (^13^C)	2 (^1^H)	2 (^13^C)
Aglycone				
2		164.39		164.66
3	6.85 *s*	103.25	6.73 *s*	103.31
4		182.10		182.03
5		161.28		161.34
6	6.39 *d* 2.2	99.25	6.39 *d* 2.1	99.29
7		162.19		162.22
8	6.76 *d* 2.2	94.44	6.73 *d* 2.1	94.34
9		157.01		157.06
10		105.52		105.57
1′		121.12		121.50
2′	7.94 *`d`* 8.8	128.66	7.43 *br*	119.25
3′	6.95 *`d`* 8.8	116.15		145.99
4′		161.50		150.14
5′	6.95 *`d`* 8.8	116.15	6.92 *d* 8.5	116.18
6′	7.94 *`d`* 8.8	128.66	7.43 *dd* 2.2, 8.5	121.36
7-*O*-glucuronopyranosyl				
1″	5.44 *d* 7.5	97.35	5.46 *d* 7.5	97.37
2″	3.52 *m*	81.01	3.53 *m*	81.07
3″	3.37 *m*	74.73	3.37 *m*	74.77
4″	3.37 *m*	71.54	3.37 *m*	71.58
5″	4.19 *d* 7.5	74.70	4.19 *d* 9.4	74.75
6″		169.10		169.15
OCH_3_	3.63 *s*	52.09	3.63 *s*	52.14
2″-*O*-*β*-glucuronopyranosyl				
1‴	4.88 *d* 8.2	101.42	4.89 *d* 8.3	101.46
2‴	4.62 *d* 8.2	73.52	4.63 *dd* 8.3, 9.6	73.58
3‴	3.47 *m*	73.56	3.48 *t* 9.6	73.61
4‴	3.36 *m*	72.02	3.36 *t* 9.6	72.08
5‴	3.85 *d* 8.2	75.37	3.85 *d* 9.6	75.42
6‴		169.17		169.22
OCH_3_	3.52 *s*	51.71	3.52 *s*	51.75
2‴-*O*-caffeoyl				
C=O		165.87		165.94
α	6.24 *d* 15.8	114.73	6.23 *d* 15.8	114.75
β	7.43 *d* 15.8	144.72	7.44 *d* 15.8	144.79
1⁗		125.86		125.89
2⁗	7.03 *d* 2.1	114.83	7.04 *d* 2.2	114.86
3⁗		145.65		145.72
4⁗		148.29		148.36
5⁗	6.76 *m*	115.82	6.77 *d* 8.1	115.87
6⁗	6.97 *dd* 2.1, 8.5	121.30	6.97 *dd* 2.2, 8.1	121.36

*s* = singlet, *d* = doublet, *t*-triplet, *dd* = double doublet, *m* = multiplet, *br* = broad. See Figure 1 for structures.

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
