# Peer review of "Acylated Flavone O-Glucuronides from the Aerial Parts of Nepeta curviflora"

_molecules, 2020, doi:10.3390/molecules25173782_

Round 1

Reviewer 1 Report

The manuscript: Acylated flavone O-glucuronides from the aerial parts of 

Nepeta curviflora. meets with the aims and scope of this journal.

The different sections are well presented and are documented with information novel and relevant, but could be useful if in the section of spectroscopy includes the mass spectra of the new compounds, this could give more support to the data provided.

Just a brief note to consider is related to the chemical composition of this species, because in the genus the presence of different kind of terpenoids are well known and responsible to many biological activities.

Reviewer 2 Report

The manuscript focused on the isolation and identification of new flavonoids in the aerial parts o Nepeta curviflora, in particular the acylated derivatives of apigenin and luteolin. In order to their identification, the Authors used following methods: high resolution mass spectrometry (positive electrospray ionization (ESI), multidimensional NMR spectroscopy as well as UV-Vis absorption spectra during a HPLC analysis). However, the most important analysis that confirms the structure of compounds is mass spectrometry. The Authors made a great effort when describing in details the results of NMR analysis (including 1H and 13C NMR spectra, COSY and HSQC experiments), however, the information describing mass spectra analysis is scarce. The gave only the values for [MH]+ ions for the studied compounds but it would improve the characteristic a lot if the Authors gave the path of fragmentation of parent ion (eg., presence of caffeoyl, glucopyranoside, apigenin and luteolin moiety – after fragmentation). What is more, ESI-MS spectra, marked as Fig.S9 and Fig. S.17 are not present in available for Reviewers in the attached supplementary word file. Also Figs. S1, S3, S6, S7 showing the results of multidimensional NMR analysis is missing. It raises doubts. Secondly, according to the Authors identification of compounds 1 and 2 as dimethyesters was caused by methylestrification of glucuronyl moieties with acidified methanol during extraction and isolation process (page 3 lines 113-115). However, compounds 1, 2 and 4 (methylester of caffeic acid) were obtained  tin the combined fractions No 25 and 26 (Table S4) (Page 4 lines 154-156), and these fraction were obtained using acetonitrile as a solvent (Tab. S4) and not using methanol. Only in the case of compounds 3,4 and 6 for which they used acidified methanol as a solvent, esterification with methanol may occur. Thus, it isn’t sure if compounds 1 and 2 reacted with methanol during isolation or they were present in this plant in natural form as methyl esters. My overall recommendation about this manuscript – reconsider after major revision.

The remaining comments are given below,

Page 1 line 20 The characterization of what? Please to precise!

Page 1 line 35 “beneficial health effects” should be changed as “beneficial pro-health effects” in this line and consistently in all the text

Page 1 line 37 “of the genus” should be changed as “of this genus”

Page 1 line 41 “antimicrobial activity of N…., dissolved in dimethylsulfoxide” should be changed as “antimicrobial activity of dimethyl sulfoxide extract of N/ menthoides” line 42 “methanolic extracts of leaves” should be changed “however, methanolic extracts of leaves”

Page 2 line 46 “secondary metabolites” should be changed as “phenolic compounds”

Page 2 line 47 “this study is thus to isolate”  should be changed as “this study was to isolate”

Page 2 line 62 “esters were most probably made” should be changed as “esters were most probably synthesized”

Page 4 line 127 “deuterio-methyl” shoul be changed as “ deutero – methyl”

Page 4 line 148 “was concentrated under reduced pressure” in order to remove methanol?

Page 4 line 150 “purified extract” or “purified dry extract”, Please to precise

Reviewer 3 Report

In this study, the authors extracted and identified a few compounds, two of which are new in Nepeta curviflora. The study is interesting. However, there are some major concerns that need to be addressed.

  1. The Mass was used in the experimental part. However, the results are not mentioned except the molecular ion for one compound. Their fragments should be analyzed as well.
  2. The supplemental figures are missing.
  3. In the biological activity part, what are the final concentrations for the tested compounds? In addition, the detailed results should be shown, since that at least XAD-7 showed positive results.
  4. This herb has a couple of functions, why only focus on its anti-bacterial activity, which appears to be minor?
  5. As stated in section 3.5, 0.5 g of NaCl in 500 ml of water (0.9% of NaCl) would not give 0.9% NaCl. It should be ~0.1%. Please double check.
  6. Which samples were subjected to preparative HPLC? Partition from Sephadex or other extract. Please clarify.
  7. Typo: (I)it embraces around 300 species. Careful proofread is needed.

Reviewer 4 Report

Acylated flavone O-glucuronides from the aerial parts of Nepeta curviflora

Corrections

These paragraphs are appropriate for discussion. They shouldn't be at the conclusion.

Previously we have reported that the free carboxyl group of glucuronyl moieties of flavonoids readily will be esterified with methanol during extraction and isolation processes involving acidified methanol as solvent [30]. This is in accord with the identification of both 1 and 2 as dimethylesters caused by methylesterfication of the two glucuronyl moieties of these  flavonoids. Small amounts of parental 1 and 2 without their methylesters were indeed detected by LC-MS analysis of Sephadex LH-20 fractions of the purified extract of N. curviflora (Fig. S10 and fig. S18). Flavonoids glycosylated with glucuronic acid have previously been reported from several Nepeta spp. [34-36].

  • Table S1 – There are Molecular mass [MH]+ data for compounds 3-6?.
  • 1H NMR spectrum of 1 (Fig. S1), HSQC (Fig. S3), HMBC (Fig. S4) and H2BC (Fig. S6) NMR spectra of 1 and 1H– 1H COSY experiment (Fig. S7), HMBC spectrum of 2 (Fig. S13) and high-resolution ESI+–MS spectrum of 2 (Fig. S17), Sephadex LH-20 fractions of the purified extract of N. curviflora (Fig. S10 and fig. S18) were not included in supplementary material.

What is Fig. S2 and S5 and others Figures.

In my opinion Table S4 and S5 : Solvent composition and elution volumes are not important.

Reviewer 5 Report

This manuscript presents the isolation and structure elucidation of two new flavonoids from Syrian catnip, along with four known phenolic compounds.  The compounds were isolated by various chromatographic techniques and characterized by spectrometric methods, chiefly multidimensional NMR.  This work adds to our understanding of the phytochemistry of the genus Nepeta.  Publication is Molecules is recommended.

There are only minor editorial corrections needed:

Line 39:  catnip [not capitalized]

Lines 103 and 184:  agar [not capitalized]

Round 2

Reviewer 3 Report

The revised version is acceptable. No more additional comments.

Reviewer 4 Report

Key words - flavone 7-O-glucuronosides correct by flavone 7-O-glucuronides

Experimental

Line 137 - mass spectrometer equipped with a triple quadrupole (QqQ configuration) mass analyser  using electrospray ionization (ESI) as detector?.

As an ionization method, electrospray ionization (ESI) is used for a CE–MS interface. Electrospray ionization mass spectrometry is a desorption ionization method.

Electrospray ionization (ESI) is a technique used in mass spectrometry to produce ions using an electrospray in which a high voltage is applied to a liquid to create an aerosol. It is especially useful in producing ions from macromolecules because it overcomes the propensity of these molecules to fragment when ionized. ESI is different from other ionization processes (e.g. matrix-assisted laser desorption/ionization (MALDI)) since it may produce multiple-charged ions, effectively extending the mass range of the analyser to accommodate the kDa-MDa orders of magnitude observed in proteins and their associated polypeptide fragments.